# Changes in sea floor productivity are crucial to understanding the impact of climate change in temperate coastal ecosystems according to a new size-based model

Asta Audzijonyte[1,2]*, Gustav W. Delius[3]*, Rick D. Stuart-Smith[1], Camilla Novaglio[1,2], Graham J. Edgar[1], Neville S. Barrett[1], Julia L. Blanchard[1,2]

**1** Institute for Marine and Antarctic Studies, University of Tasmania, Hobart, Australia, **2** Centre for Marine Socioecology, University of Tasmania, Hobart, Australia, **3** Department of Mathematics, University of York, York, United Kingdom

* asta.audzijonyte@utas.edu.au (AA), gustav.delius@york.ac.uk (GWD)

**Data Availability Statement:** All code and data are available at https://github.com/astaaudzi/SEAmodel

## Abstract

The multifaceted effects of climate change on physical and biogeochemical processes are rapidly altering marine ecosystems but often are considered in isolation, leaving our understanding of interactions between these drivers of ecosystem change relatively poor. This is particularly true for shallow coastal ecosystems, which are fuelled by a combination of distinct pelagic and benthic energy pathways that may respond to climate change in fundamentally distinct ways. The fish production supported by these systems is likely to be impacted by climate change differently to those of offshore and shelf ecosystems, which have relatively simpler food webs and mostly lack benthic primary production sources. We developed a novel, multispecies size spectrum model for shallow coastal reefs, specifically designed to simulate potential interactive outcomes of changing benthic and pelagic energy inputs and temperatures and calculate the relative importance of these variables for the fish community. Our model, calibrated using field data from an extensive temperate reef monitoring program, predicts that changes in resource levels will have much stronger impacts on fish biomass and yields than changes driven by physiological responses to temperature. Under increased plankton abundance, species in all fish trophic groups were predicted to increase in biomass, average size, and yields. By contrast, changes in benthic resources produced variable responses across fish trophic groups. Increased benthic resources led to increasing benthivorous and piscivorous fish biomasses, yields, and mean body sizes, but biomass decreases among herbivore and planktivore species. When resource changes were combined with warming seas, physiological responses generally decreased species' biomass and yields. Our results suggest that understanding changes in benthic production and its implications for coastal fisheries should be a priority research area. Our modified size spectrum model provides a framework for further study of benthic and pelagic energy pathways that can be easily adapted to other ecosystems.

with an archival release at https://doi.org/10.5281/zenodo.8281030.

**Funding:** This study was supported by the ARC Discovery grant DP170104240 (https://www.arc.gov.au/) (JLB, RDSS) and Pew Fellows Program in Marine Conservation (https://www.pewtrusts.org/en/projects/marine-fellows) (AA). The funders had no role in study design, data collection and analysis, decision to publish, or preparation of the manuscript.

**Competing interests:** The authors have declared that no competing interests exist.

**Abbreviations:** AIC, Akaike information criterion; ATRC, Australian Temperate Reef Collaboration; MSSS, multispecies size spectrum; RLS, Reef Life Survey.

## Introduction

Climate change is causing a rapid and poorly understood reorganisation of natural ecosystems worldwide [1,2], with implications for human well-being and conservation goals [3,4]. Shifts in species distributions, altered primary production, and changing physiological rates have been documented in terrestrial and aquatic ecosystems and are reshaping food webs [5–7]. Understanding, predicting, and mitigating consequences of these changes on food production and biodiversity conservation represent important research priorities [8], but predictions remain extremely difficult due to the complexity of interactions [9]. One of the main challenges in attempting to predict responses of the valuable fishes in marine ecosystems is that changing ocean climates can impact fish populations through multiple direct and indirect mechanisms. These include changes in primary productivity, the size distributions of their food resources, fish community composition and size structure, and the influences of temperature on growth and other metabolic processes [10,11]. Through the long-established principles of bioenergetics, we expect that changes in food availability (ultimately through changes in primary producers at the base of food webs) and temperature will both affect growth and size structure of fish populations [12]. These changes, in turn, through size-based food web interactions are expected to alter mortality and reproduction, affecting population abundance and community structure [13]. To understand how these changes will affect food webs, we need tools that can resolve physiological processes, body size structures, and species interactions dynamically, in a robust and computationally efficient way.

Physiologically structured food web models are particularly useful for this purpose because they incorporate key organismal processes that likely respond to temperature, resolve size structure interactions between organisms, and allow species abundance, interactions, growth, and yields to emerge dynamically from the assumptions and knowledge about species' life-history, density dependence, and spatial overlap [13,14]. Such modelling tools have already been applied to explore climate change impacts on marine ecosystems at regional and global levels and have predicted declining fisheries yields, decreasing energy transfer efficiency, decreasing growth rates, and smaller fish body sizes with warming [8,10,15–17]. Regardless of these advances, climate and productivity change impacts remain highly uncertain, especially for coastal regions, where the relative contributions of pelagic versus benthic primary production are largely unknown [18,19]. An additional consideration is that fish populations are affected both by overall productivity shifts and by changing food resource size composition [20].

In most food web models used to explore ecological impacts of climate change under future climate scenarios, projected productivity and temperature changes have been applied together, typically by forcing these models with temperature and production time series generated by Earth System Models (e.g., Coupled Model Intercomparison Project), and concentrating on long-term projections, summarising results across the entire community [16,21]. While needed for large-scale predictions, this approach makes it hard to understand the relative importance and uncertainties associated with production, temperature, size structure, and their interactions. An alternative approach is to examine productivity, resource size structure, and temperature treatments separately and combined and in this way better understand the mechanisms and interactions driving observed ecosystem changes [22].

A major challenge for predictions in coastal ecosystems is that global Earth Systems Models do not yet resolve benthic primary production pathways that are likely to play important roles for coastal communities, and these models are also poor at resolving pelagic primary production in the narrow strip of the ocean close to the coast [23]. Even though global ecological models that capture ecological interactions between simplified benthic and pelagic pathways do exist for climate projections [8,24,25], they are focused on shelf areas. They also assume

that benthic production is detrital and dependent on plankton subsidies, rather than independent primary production pathway, as is the case in shallow water systems. Yet, shallow water marine ecosystems are undergoing some of the most rapid and accelerating changes due to human impacts [26,27], provide livelihood for over 1 billion people, and harbour most of marine biodiversity. For example, coral reefs alone are estimated to include approximately 25% of all marine species [28].

Shallow water systems differ from open ocean in several important ways, and models developed for open ocean systems may be limited in their capacity to make predictions for shallow ecosystems. First, light penetrates to the substratum in shallow coastal ecosystems, supporting an additional primary production pathway through benthic producers. The benthic production pathway encompasses production by microphytobenthos, tiny turf algae, and macroalgae up to giant kelp or a diverse array of corals, fuelling a multitude of benthic invertebrate consumers and predators [29,30]. The relative contributions of benthic and pelagic pathways in coastal ecosystems remains debated [31] and likely varies in time and space, yet it seems reasonable to expect that climate change will impact these 2 pathways in different ways. Second, higher structural complexity in coastal ecosystems represents an important difference to open ocean and shelf ecosystems, influencing the transferability of model-based predictions. The habitat and resource complexity provided by corals and kelps growing on rocky and coral reefs provides important refugia from predation [32] and supports higher functional diversity of reef trophic groups able to capitalise on the diverse range of resources [33]. Unlike in open ocean or shelf systems [13,25,34], many fishes on shallow reefs feed on low trophic level resources, rather than on each other, which suggests that changes in resource abundance or size structure might disproportionately impact the fish community [35].

One feasible approach to better understanding climate change responses in coastal systems is thus to extend physiologically structured food web models developed for offshore systems, to better include the features of coastal ecosystems that make them different to pelagic and shelf systems. For this purpose, we outline a model parameterised and calibrated for a well-studied temperate rocky coastal reef system in a climate change hotspot (Fig 1), where most of the trophic groups, including some large fish species, rely on benthic resources (Fig 2). Using the model, we attempt to unpack complex ecosystem responses, specifically focussing on evaluating changes in coastal fish communities that can arise from varying scenarios of change in pelagic and benthic resources (abundance and size composition) and physiological fish responses to temperature. We ask 2 broad sets of questions: (1) Do changes in the pelagic and benthic production pathways have similar or opposing effects on biomass, average body size, and yields of reef fishes? (2) How do fish species and community-level responses to plankton and benthos changes compare to their physiological responses to warming, and do temperature-driven physiological changes amplify or dampen warming impacts from food resource availability?

## Methods

### Model system and selection of taxa

To study the impacts of global warming on coastal fish communities, we focused on Tasmanian reefs within the SE Australian climate change hotspot, parameterising a multispecies size spectrum (MSSS) model for a well-studied temperate rocky reef community (Fig 1). These communities have been monitored regularly for the last 30 years using standardised quantitative underwater visual surveys undertaken as part of the Australian Temperate Reef Collaboration (ATRC) monitoring program conducted since 1992 [36], and Reef Life Survey (RLS) program since 2008 [37,38], as described in an online methods manual at http://www. reeflifesurvey.com. In this study, we focus on 4 long-term monitored locations, all positioned

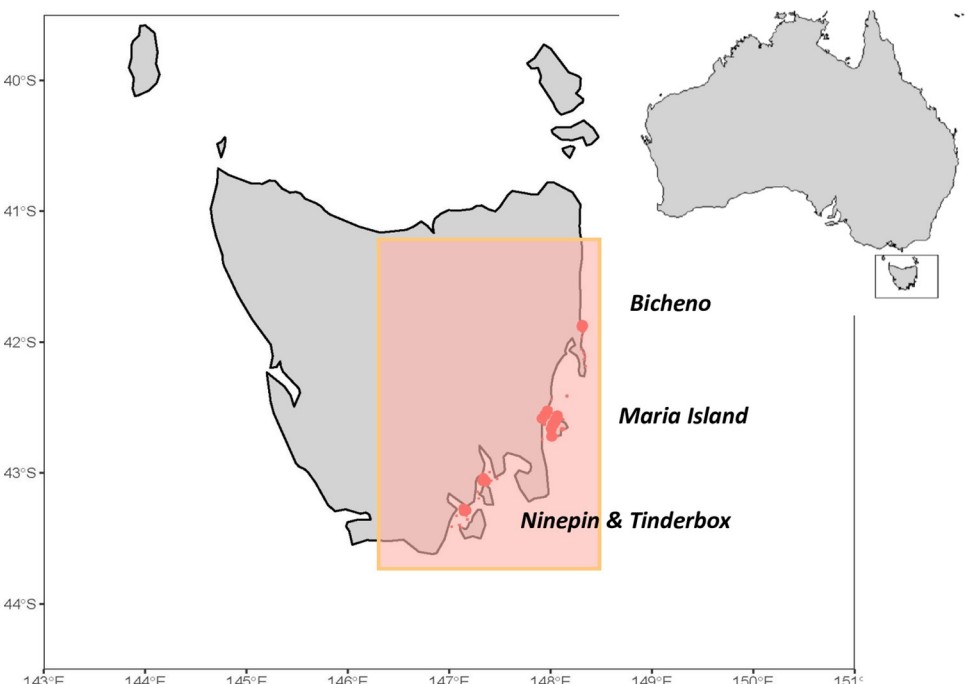

**Fig 1. Area of the model domain in Tasmania (shaded in orange) and survey locations used to calibrate the model (circle area proportional to the number of surveys).** Public domain map from naturalearthdata.com.

within a single biogeographical province. Data from the 1990s were used to parameterise the model, whereas data from 2000s and 2010s were used as an indication of empirically observed ecosystem changes versus those simulated in alternative model scenarios (Fig G in S1 Text).

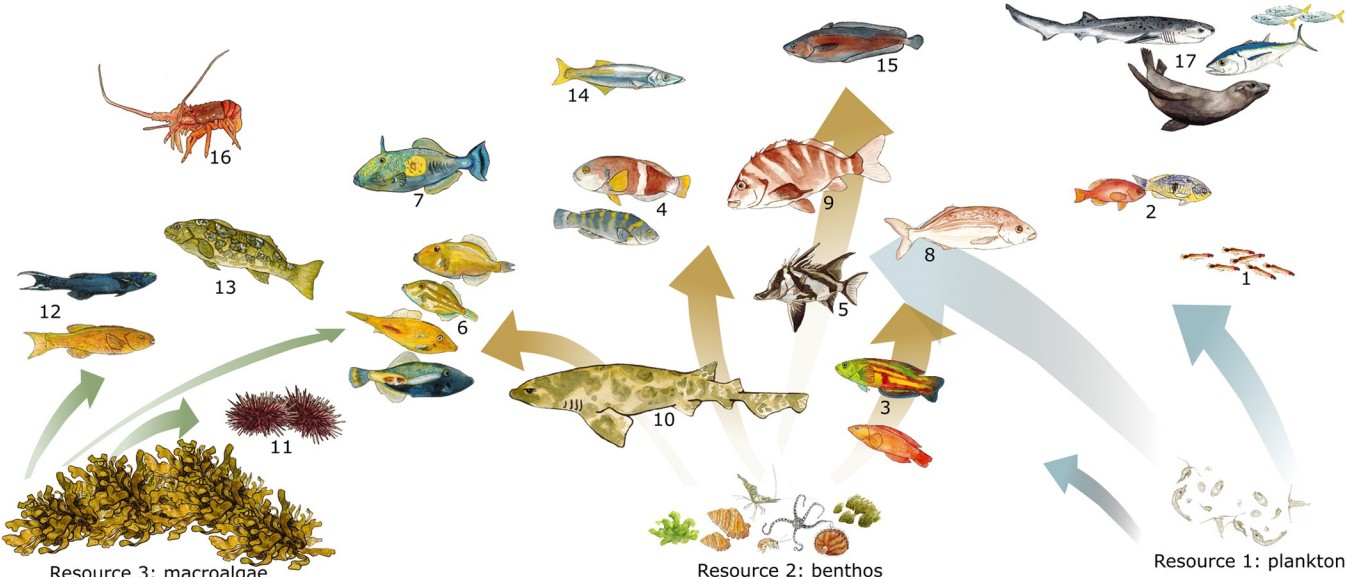

**Fig 2. Schematic illustration of the model groups and 3 background resources used in this study.** Detailed feeding interactions across species are not shown; coloured arrows indicate only the dominant feedings pathways for herbivores (left, from macroalgae), benthivores (middle, from benthos), and planktivores (right, from plankton). All species feed on plankton in earliest life stages (indicated with the large arrow). Predators (#14–17) are shown on the top of the figure, but for other groups, position in the figure does not reflect their trophic level. The image was drawn by Amy Coghlan.

**Table 1. List of model species and groups assigned into 4 main trophic categories, approximate maximum body size, and modelled fishing mortality.**

| Common name | Scientific name | ~$w_\infty$ (kg) | F (year$^{-1}$) |
|---|---|---|---|
| *Planktivores*<br>Hulafish (1) | *Trachinops caudimaculatus* | 0.04 | 0 |
| Barber Perch (2) | *Caesioperca rasor* | 0.62 | 0.05 |
| *Benthivores* | | | |
| Senator wrasse (3) | *Pictilabrus laticlavius* | 0.40 | 0.05 |
| Purple and bluethroat wrasse (4) | *Notolabrus tetricus* and *Notolabrus fucicola* | 1.60 | 0.15 |
| Long-snouted boarfish (5) | *Pentaceropsis recurvirostris* | 2.20 | 0.15 |
| Leatherjackets (6) | *Acanthaluteres vittiger* and *Meuschenia australis* | 3.00 | 0.05 |
| Six-spine Leatherjacket (7) | *Meuschenia freycineti* | 4.30 | 0.15 |
| Bastard Trumpeter (8) | *Latridopsis forsteri* | 5.20 | 0.15 |
| Banded Morwong (9) | *Cheilodactylus spectabilis* | 13.50 | 0.15 |
| Draughtboard Shark (10) | *Cephaloscyllium laticeps* | 16.00 | 0.10 |
| *Herbivores* | | | |
| Urchins (11) | *Heliocidaris*, *Centrostephanus*, and *Goniocidaris* | 0.35 | 0.15 |
| Herring Cale (12) | *Olisthops cyanomelas* | 3.40 | 0.15 |
| Marblefish (13) | *Aplodactylus arctidens* | 4.40 | 0.15 |
| *Predators* | | | |
| Long-fin Pike (14) | *Dinolestes lewini* | 1.00 | 0.15 |
| Red Cod (15) | *Pseudophycis palmata* | 1.50 | 0.15 |
| Rock lobsters (16) | *Jasus edwardsii* | 3.00 | 0.15 |
| Predators (general) (17)* | Mammals, birds, sharks, tunas, and other fish | 5.00 | 0.10 |

Further details are given in S1 Text section 1.2.

*The biomass of predators is not taken from surveys but assumed here to represent average predation from all large predators. Note that fishing mortality in a continuous size-based model, such as the one used here, is not directly comparable to mortality in age-structured stock assessment models. Numbers next to common names refer to numbers in Fig 2.

Selection of functional groups for inclusion in the model can have large impacts on the predicted ecosystem responses [39]. To minimise the subjective choice of model groups, we selected all species that satisfied the occurrence or biomass thresholds (see S1 Text section 1.2) in standard fish surveys (500 m$^2$ area; see [40] for survey methods) and pooled species with similar life-history characteristics and diets. The final list comprised 14 fish and shark species and species groups (Table 1 and Fig 2), which together accounted for over 90% of average observed biomass per survey in the 1990s. To account for the fact that abundance of mobile larger predators (marine mammals, tunas, mobile sharks, pelagic predatory fishes, birds) may not be correctly reflected in underwater visual surveys on reef habitats, we did not use survey data for predator abundance but included a general large predator species to represent the unaccounted predation. This general predator was assumed to grow to a maximum of 5 kg, which is an approximate average across birds, commonly observed mackerel and other medium sized pelagic fish species (maximum size of 2 to 3 kg), but also occasional and much larger sharks and mammals. The predator abundance was chosen based on general expert knowledge, and we also assessed sensitivity of model outcomes to this assumption by removing the predator completely and repeating the simulations (S1 Text section 2.4). Most of the smaller sized invertebrate species were represented through the benthic background resource, but 2 ecologically and economically important groups (urchins and rock lobsters) were modelled explicitly as dynamic size structured groups. Urchins and lobsters were included as separate functional groups because they are very abundant in the ecosystem (urchins are the most

abundant group; see Table A in S1 Text), they play important role as herbivores and predators, which would not be resolved if these groups were treated as a part of benthic resource spectrum, and because they reach sizes (Table 1) that are considerably larger than the maximum size of the benthic resource spectrum (5 g; Table B in S1 Text). Further details on the selection of model groups are presented in S1 Text section 1.2 and Table C.

## General physiologically structured size spectrum modelling framework

To explore how temperature effects on fish physiology compare and interact with changes in the pelagic and benthic resource abundance and size composition, we use an MSSS modelling framework [13] and its implementation in the R package *mizer* [41] (sizespectrum.org/mizer). The framework has been used in several recent studies [34,42–44], and its theoretical basis is described in detail on sizespectrum.org, *mizer* vignette, and [13]. Here, we only briefly mention the key assumptions, with more details provided in S1 Text section 1.1 and Tables A and B in S1 Text.

The *mizer* application of the MSSS model simulates a dynamic size-structured background resource and user-defined size structured groups (fish and invertebrates), which feed on the background and on each other. The groups can either represent a single species or groups of species and are referred to here as model groups. The dynamics of size structured groups is summarised by the McKendrik–von Foerster equation, where change in abundance at size through time depends on emergent somatic growth $g_i(w)$ [g year$^{-1}$] and mortality $\mu_i(w)$ [year$^{-1}$] of that size class. In this way, size-specific growth and mortality determine how many individuals enter and leave each size class:

$$\frac{\partial N_i(w)}{\partial t} + \frac{\partial g_i(w)N_i(w)}{\partial w} = -\mu_i(w)N_i(w).$$ (1)

Growth depends on the availability of food, Holling type II feeding response, assimilation efficiency, maintenance costs, allocation to reproduction, and growth efficiency (applied here but typically not used in other *mizer* applications). Feeding and food availability are strongly determined by size so that each size group feeds on the food sizes available to it (determined by the predator prey mass ratio kernel of each species, as well as species and resource interaction matrix). Allocation to reproduction is also size specific, determined by the weight at 25% and 50% maturation (at the population level). Size classes at 50% maturation allocate 50% of their net energy to reproduction and the rest to growth. This allocation to reproduction increases in larger size classes, reaching 100% in the largest size class. Mortality includes constant background mortality, predation, starvation, senescence, and fishing mortality. Predation, senescence, and fishing mortalities are size dependent, as described in S1 Text section 1.1. The minimum, maximum, and maturation sizes for each size-structured group are set by the user (Table 1 and Tables B and C in S1 Text). The numbers in the first size class are determined by the continuous density-dependent recruitment (see below). The numbers of individuals in each size class are an emergent property of the model. These numbers depend on the growth and mortality in each size class, which, in turn, depends on the food availability and growth, predation, fishing mortality, and other mortality sources. Therefore, biomasses, yields, and average sizes are also an emergent model property, determined by size specific growth and mortality. For example, higher resource abundance means that individuals grow faster, but it could also lead to increased predation rates. Depending on the relative change in growth and mortality on each size class, increased resource abundance could lead to either increasing or decreasing average sizes of the fish species. Similarly, while the background mortality rate is constant across size classes, different growth rates across the size classes mean that, in some

size classes, individuals stay longer than in others, which, in turn, affects the final number of individuals affected by background mortality in a specific size class.

The temporal dynamics of the background resource spectrum ($N_R$) is modelled through semi-chemostat dynamics [14] and depends on the emergent predation mortality ($\mu$), resource regeneration rate ($r_o$), and resource carrying capacity ($\kappa$) scaled by the size spectrum slope of the resource ($\lambda$):

$$\frac{\partial N_R(w, t)}{\partial t} = r_o w^{n-1} \left( \kappa w^{-\lambda}(w) - N_R(w, t) \right) - \mu(w) N_R(w, t) \qquad (2)$$

where $n$ is the maximum food intake body scaling exponent of size structured groups (see [45] for further derivations of background spectra dynamics). Further details about resource parameters are provided in the S1 Text sections 1.7 and 1.8.

## Modification of the modelling framework for coastal ecosystems: Benthic and pelagic production pathways

In this study, we modified the standard MSSS model framework to suit coastal ecosystems by introducing 3 size-structured background spectra to represent pelagic, benthic (turf and invertebrates), and macroalgal (kelp, seaweed) resources. This distinction is important for shallow water communities, because (1) a number of fish species are specialised to feed on either pelagic, benthic, or macroalgal food and feeding on both benthic and pelagic food resources is size based [46]; (2) in contrast to pelagic ecosystems, benthic primary and secondary production are likely to provide substantial energy inputs, independent from pelagic production [30,47]; (3) size spectrum slopes of pelagic and benthic producers, and their responses to climatic warming, are likely to differ; and (4) many fish species have ontogenetic diet shifts, starting with pelagic (planktonic) prey and switching to benthic prey as they grow [42]. If pelagic and benthic resources respond differently to warming seas, the consequences for fishes and communities are likely to differ from those based on the assumption of a fixed resource for each species.

Ontogenetic diet shifts should ideally emerge dynamically in the model from assumptions about size preferences, habitat preference, and relative food abundances, yet their reproduction in physiologically structured models can be challenging [42,48]. In this study, we aimed to reproduce emergent diet shifts by assuming different size ranges and size abundance slopes for the 3 resource spectra, and by specifying availability of each spectrum to each model species, implicitly representing habitat and food preference (pelagic, benthic, and macroalgal). In this way, each background resource has specific parameters for carrying capacity ($\kappa$), size spectrum slope ($\lambda$), and regeneration rate ($r_o$), as well as minimum and maximum size of the spectrum (Table 2), whereas each species has resource-specific vectors indicating maximum availability of each resource (see below and S1 Text section 1.4, Table D and Fig B in S1 Text). This means that emergent ontogenetic shifts could be reproduced by just 3 species-specific parameters, identifying preference for each of the 3 background resources. This modification of the *mizer* package, allowing multiple size-structured background spectra and species-specific preferences for different resources, is now available as a *mizerMR* extension to the main *mizer* package (https://github.com/sizespectrum/mizerMR).

## Climate change scenarios: Changes in plankton and benthos abundance and size

To assess species and ecosystem responses to climate change scenarios, we explored 9 alternative benthic and pelagic resource change scenarios (see below and Table 2) fully crossed with

**Table 2. Scenarios of plankton and benthos resource changes.** Each of these scenarios was combined with warming and fishing scenarios in a fully crossed ANOVA-style design.

| Resource change scenario name | Plankton spectrum | | Benthos spectrum | |
|---|---|---|---|---|
| | $\lambda_P$ | $\kappa_P$ | $\lambda_B$ | $\kappa_B$ |
| 1. Baseline | −2.15 | 2 | −1.9 | 6 |
| *Plankton abundance change* | | | | |
| 2. More plankton | −2.15 | **2.6** | −1.9 | 6 |
| 3. Less plankton | −2.15 | **1.5** | −1.9 | 6 |
| *Plankton size structure change* | | | | |
| 4. Small plankton | **−2.18** | 2 | −1.9 | 6 |
| 5. Large plankton | **−2.12** | 2 | −1.9 | 6 |
| *Benthos abundance change* | | | | |
| 6. More benthos | −2.15 | 2 | −1.9 | **9** |
| 7. Less benthos | −2.15 | 2 | −1.9 | **4** |
| *Benthos size structure change* | | | | |
| 8. Small benthos | −2.15 | 2 | **−2.0** | 6 |
| 9. Large benthos | −2.15 | 2 | **−1.8** | 6 |

physiological temperature responses (i.e., temperature impacts on vital physiological rates) in an ANOVA-style design. For all scenarios, low fishing mortality was assumed (Table 1). This gave a total of 18 scenarios, each run with 29 alternative parameter combinations to assess model output uncertainty (see below), resulting in 522 simulation outputs. Each scenario was initiated from baseline equilibrium conditions (no resource or temperature change, main parameter set), then new conditions imposed instantaneously and applied for 150 years of the model run. In nearly all cases, simulations settled into a new stable or oscillating equilibrium within 20 to 40 years. For the analyses here, we did not focus on the transient dynamics but compared final equilibrium conditions to the initial equilibrium state. To assess the impact of parameter uncertainty on predicted ecosystem responses in each of the 18 climate change impact scenarios, we compared the differences between baseline and specific scenarios for each of the 29 parameter combinations, respectively. This means that we were not focused on the total amount of uncertainty that combines both parameter and scenario uncertainty, but on whether a similar change in the modelled community occurred between the baseline and a specific model scenario, regardless of parameter combination.

## Resource parameters in baseline and climate change scenarios

For many marine ecosystems, the slope of the plankton normalised abundance spectrum varies between −2 and −2.2 [13,49]. For low productivity waters, such as SE Australia, slopes are often steeper, and, therefore, for the baseline scenario, we use the value of −2.15 (see S1 Text sections 1.7 and 1.8 for further details on background parameter values). Slopes and abundances of the benthos spectrum for the baseline scenario were estimated from empirical data along SE Australian coast (Figs D and E in S1 Text). The macroalgal spectrum was modelled here as size structured background resource for simplicity, even though herbivore feeding on macroalgae is unlikely to be primarily size based. To ensure high abundance of the macroalgal resource across various size groups, we assumed relatively flat slopes (−1.6) and high abundance (16 g/m²). These values do not account for the large kelp stands in many rocky reef communities, but entire kelp is not typically consumed by the local herbivores. The selection of macroalgal resource parameters mostly aimed to ensure high feeding levels in herbivorous species (i.e., no food limitation).

The goal of our study was to explore the multispecies coastal community responses to warming driven changes in pelagic and benthic resource abundance and size structure independent of, and interacting with, changes in temperature. Although the model is parameterised for a specific ecosystem, it has a broader goal of investigating the relative importance of physiological temperature-driven changes versus food availability in a temperate coastal fish community with multiple primary production pathways. We were not aiming to replicate specific climate change scenarios because of large uncertainties in forecast changes for SE Australia and coastal ecosystems in general [17,21]. For example, it is generally expected that increasing temperature favours smaller body size and that higher temperatures will lead to steeper background resource size spectra [21,50–52] (Fig C in S1 Text). Yet, empirical evidence shows that changes in temperature also affect nutrient availability and grazing pressure [53,54], that benthic resource size spectra may be insensitive to temperature [53,55], and that field data from the last 10 years show increasing zooplankton abundance off Maria Island (S1 Text section 1.7). We therefore explored a range of resource change scenarios, with plankton or benthos abundance ($\kappa$) and slopes ($\lambda$) increasing or decreasing independently (Table 2). We assumed that plankton or benthos $\kappa$ values increase or decrease by approximately 30%, and slopes for plankton can change by 0.03 and for benthos by 0.1 (see S1 Text sections 1.7 and 1.8). Note that changes in the resource slope ($\lambda$ parameter) also affect the overall resource abundance below and above the size of the pivot point. By default, the pivot point in *mizer* is set at 1 g, so steeper slopes will increase plankton abundance, but mostly so at smaller sizes (since maximum plankton size is 1 g). Steeper slopes for the benthic resource will increase the abundance of small benthos organisms (<1 g) and decrease the abundance of organisms >1 g. Scenarios with resource slope changes produced generally similar results to those with resource abundance changes, and their results are presented in Fig G in S1 Text.

## Physiological response in climate change scenarios

To model temperature effects on physiology in a manner consistent with other similar physiologically structured models, we followed the metabolic theory of ecology approach [56] and assumed that temperature affects the rate of metabolism, search rate, maximum food intake rate, and background and senescence mortality [57–59]. All rates were assumed to increase exponentially with temperature based on Arrhenius temperature correction factor

$$r(T) = e^{\frac{A_r(T-T_{ref})}{kTT_{ref}}} \tag{3}$$

where $A_r$ is the activation energy [eV] for individual rate $r$ here assumed to be 0.63 (see Lindmark and colleagues [60] for a review of activation energies in different fish species), $T$ is temperature [K], $T_{ref}$ is the reference temperature where temperature scaling equals 1 (here assumed to be 12°C or here 285.27 K), and $k$ is Boltzmann's constant in eV K$^{-1}$ (8.617×10$^{-5}$ eV K$^{-1}$). This representation of temperature is a simplification (see Discussion) and ignores potential differences across rates, species, and time. Moreover, we do not explore temporal evolution of the model but just equilibrium conditions under alternative physiological and food availability states. This assumption of identical responses across species allowed us to assess how species interactions might modify species biomass and mean size changes due to the physiological response alone and enabled a more consistent comparison with other models exploring climate change effects [8,16,22]. The simple physiological response assumption also gave us a tractable number of simulations that could be analysed in a 3-way ANOVA framework (resource: physiology: trophic group identity; see the Comparing alternative scenario outputs section below). As with productivity, changes in temperature-driven physiological rates were applied instantaneously to the baseline equilibrium conditions and the model run

to a new equilibrium to assess species responses. We follow predictions of RCP8.5 scenario, forecasting temperature increase of ca. 2.5°C by 2100 for SE Australia (S1 Text sections 1.7–1.8); hence, temperature increase scenarios were run at 14.5°C temperature, compared to 12°C reference temperature. The selected parameter values lead to ca. 20% increase in physiological rates.

## Species parameters and model parameterisation

Species parameters were either selected based on general size spectrum model assumptions [13,45] or estimated for this model (S1 Text section 1.3). For many rocky reef species, maturation sizes and growth curve parameters are not known; hence, we could not use standard growth-based approaches to estimate intake and metabolism parameters [13,34] but instead estimated them from general interspecific relationships [13,61] or derived them from broadscale species correlations and estimates used in the Dynamic Energy Budget database (see S1 Text section 1.3, Table D, Fig A in S1 Text, and https://github.com/astaaudzi/SEAmodel code for full details). To our knowledge, this is the first study providing alternative multispecies parameter derivations for MSSS models and, thus, a novel test example for new model developments in data-poor regions. Parameters defining intake rates were further explored in the uncertainty evaluation framework (below).

In addition to species-specific life-history and physiological parameters, food web models are also highly sensitive to parameters defining species interactions. In MSSS models, the emergent consumption of a species or a background resource depends on its relative abundance at size, the consumer predation kernel (Table A in S1 Text, Eq3) that sets limits on the size ranges that each species can eat, and the species interaction matrix (Table A in S1 Text, Eq14) that defines the maximum proportion of a prey species or resource biomass available for consumption at each time step. Predation kernel parameters were selected to reflect general expert knowledge and evidence from earlier studies [34,62] (Table D in S1 Text). For resource consumption, the availability of each background resource for each species was set through species- and resource-specific scalars (Table D in S1 Text), selected on the basis of general knowledge about functional groups (e.g., planktivores, herbivores, invertivores, and predators [33]), and aimed to achieve realistic emergent species diets (Fig B in S1 Text). For species interactions, we aimed to reduce the interaction matrix to key parameters. Briefly, for the 17 species, the interaction matrix can have up to 17 × 17 parameters, defining specific predation preferences for each species pair (Table E in S1 Text). In spatially implicit models, such as the one assumed here, the interaction matrix can be complex and used to model species physical overlap [34] or detailed diet preferences [42]. In this study, where species have full spatial overlap, the interaction matrix was reduced to just 5 parameters aimed to reflect general diet preferences and vulnerability to predation (Table E in S1 Text). Two parameters (0 and 0.7) indicate absence or presence of predation, and the remaining 3 parameters were used to adjust vulnerability to predation through antipredatory behaviour, e.g., schooling in small bodied species, or morphological defences in urchins (S1 Text section 1.4). The interaction matrix parameters were further explored in uncertainty analyses (see Methods below).

One of the key parts of model parametrisation and calibration is finding parameters for reproduction and recruitment that enable species coexistence, reasonable biomass values, and expected resilience to exploitation [63]. To this end, we iteratively tuned 2 species-specific reproduction parameters—the maximum recruitment parameter $R_{max}$ and the reproductive efficiency parameter $\varepsilon$ (Eq10 and Eq11 in Table A in S1 Text). The $R_{max}$ sets the upper recruitment limit and represents density-dependent processes that are not dependent on the biomass, such as habitat availability or disease. The parameter $\varepsilon$ determines the proportion of total

reproduction energy converted into egg biomass and is used to account for reproduction inefficiencies, costs, and early egg mortality [13]. In this way, $\varepsilon$ allows a linear relationship between stock biomass and recruitment, whereas $R_{max}$ adds the nonlinear density dependence on recruitment. The 2 parameters were tuned to satisfy 2 following conditions. First, $R_{max}$ was adjusted to ensure that relative model biomasses for each species at equilibrium conditions were within 20% of the observed relative biomasses in visual surveys from the 1990s (across sites and years; Table C in S1 Text); this also ensured species coexistence. Second, $\varepsilon$ was adjusted so that individual species' vulnerability to fishing and fishing mortality at maximum yield ($F_{msy}$) was within the expected range given species body size and life-history traits [13] (S1 Text section 1.5 and https://github.com/astaaudzi/SEAmodel, code archival release at https://zenodo.org/badge/latestdoi/220103456).

## Parameter uncertainty evaluation

The parameter selection procedure briefly introduced above and described extensively in the S1 Text section 1.6 leads to 1 set of parameters that generates realistic model behaviour. Most multispecies food web and complex models with large numbers of parameter apply a similar procedure [63–66], with more rigorous parameter uncertainty evaluation limited to specific cases [43,67] and mostly focusing on species recruitment parameters ($R_{max}$) evaluated against time series of catches. In this study, we conducted uncertainty evaluation of 37 parameters, focusing on recruitment and species interaction parameters that were found to be influential and highly uncertain during the model exploration: species-specific $R_{max,i}$ and search rates ($\gamma_i$) and 3 species interaction matrix parameter defining vulnerability to predation (S1 Text section 1.6).

To apply a novel but relatively straightforward uncertainty evaluation procedure that could be replicated in other MSSS models, we used a rejection algorithm, similar to the Approximate Bayesian Computation approach [68]. For this, we ran baseline scenario simulations with many parameter combinations and rejected simulations that did not satisfy specific criteria on emergent community properties, given the ecological and biological knowledge about the system [69]. To do this, for the 37 explored parameters, we generated $1.8 \times 10^6$ possible parameter combinations by sampling these 37 parameters from 2 times larger space for each parameter. Specifically, for each of these runs, the vector of the 37 parameters was multiplied by a random uniform vector ranging from 0.5 to 2 so that each parameter could take any random value that was up to 2 times smaller or larger of its original value. A vast majority of these parameter combinations produced unrealistic model outcomes in terms of species relative abundances (species went extinct). Therefore, to explore the smaller space around the original parameter values, we additionally sampled $0.4 \times 10^6$ parameter combinations, but this time multiplying the original parameter vector by a random uniform vector ranging from 0.8 to 1.2 (i.e., parameter values were allowed to vary by 20%). Both of these parameter resampling procedures results in a total of potential $2.2 \times 10^6$ parameter combinations (somewhat similar to MCMC sampling in a Bayesian analysis). We then used parallel computing to run the baseline simulation with each of these combinations. The model was run for 150 years to ensure that equilibrium conditions were achieved, but we used a time step of 0.5 years (instead of 0.2 years) to speed up the calculations and started each run using equilibrium model state from the initial "tuned" set of parameters. Each run was rejected if, at the end of the 150-year simulation, biomasses of at least 1 species were 4 times smaller or larger than the average biomasses observed in underwater surveys. We allowed average biomasses to range 4 times because across year variation in underwater surveys also showed similar levels of variability (Fig A in S1 Text). Allowing biomass to vary extensively also better reflects the fact that natural ecosystems are never truly in an equilibrium state and species biomasses naturally fluctuate. Our alternative

parameter combinations therefore reflect a range of possible ecosystem states. This criterion alone reduced the parameter space from $2.2 \times 10^6$ to $287 \times 10^3$ combinations. The remaining parameter values were used in the second step, where for each parameter combination, we ran the simulation 5 times, saving additional model outputs to explore emergent ecosystem responses and imposing additional mortality on key model species (set as fishing mortality). These "extra mortality" runs included (1) no extra mortality; (2) extra mortality on urchins; (3) extra mortality on "predator"; (4) extra mortality on lobsters; and (5) extra mortality on the planktivore *Trachinops caudimaculatus*. These runs were assessed using 7 additional criteria, describing general characteristics of an unfished system (feeding level, diets) or the expected ecosystem's response to imposing extra mortality on key species (S1 Text section 1.6). Once all the criteria were applied, we were left with a set of 28 parameter combinations, which, together with the initial set of parameters, gave the final set of 29 parameter combinations. These were used to explore uncertainty around alternative scenarios. We emphasise that our protocol does not adequately explore full parameter space and cannot be treated as probabilistic statistical uncertainty evaluation; hence, we do not use confidence intervals or posterior probability terms. However, it provides some understanding of variation in model outputs and system responses to perturbations under different plausible parameter values.

## Comparing alternative scenario outputs

Baseline model dynamics and emergent properties were compared to available data and knowledge on species biomasses, growth, and emergent diets (S1 Text sections 1.4, 2.1, Figs A and B in S1 Text). For each scenario, we calculated 4 characteristics for all model groups—biomasses, yields under constant fishing mortality (Table 1), and mean body weight for all individuals above 2 cm in length. All response variables were assessed at equilibrium conditions (100 years after applying new resource and temperature parameters). Because some scenarios settled into oscillating equilibrium with ca. 10-year periodicity, statistics were calculated as averages of the last 30 years.

As individual species within each of the 4 trophic groups showed similar responses to projected resource changes, we assessed more general trophic group-level responses using mixed-effect ANOVA analyses. Separate analyses were conducted for plankton and benthos slope ($\lambda$) and abundance ($\kappa$) scenarios, making 4 sets of analyses in total. For each analysis, variation between species was treated as a random effect, whereas resource change ($Re$) (-1, 0, 1), warming ($W$) (0, 1), and trophic group ($T$) were fixed effects. We explored different levels of interactions, starting from the full model with a 3-way interaction, then reducing the complexity to include all 2-way interactions *response ~ Re\*W + W\*T + Re\*T + (1|species)*. The 2-way model was typically selected as the best model using Akaike information criterion (AIC) (Table G in S1 Text), although the AIC score was used for indication only, because simulation data are not strictly suitable for formal statistical tests. The ANOVA analyses conducted in this study are used to explore and summarise the direction and magnitude of effects and interactions, and we do not focus on *p*-values or strict hypothesis testing [70]. Statistical analyses were conducted using R version 4.0.0 [71], the "effects" (v4.1–4; [72]), and the "emmeans" (v1.4.7; [73]). To illustrate changes in species biomasses and mean body size that occurred in the modelled ecosystem over the last 2 decades, we plotted empirically observed change in mean biomass per survey (from 1992 to 2018) and estimated changes in mean body size from Bayesian mixed-effect modelling analyses [74]. These empirically observed changes are only provided for illustrative purposes (Fig G in S1 Text), because our modelled scenarios aim to explore interactions between different climate change processes rather than reflect realistic changes in the ecosystem.

## Results

### Model parameterisation in the baseline scenario

Before applying climate change scenarios, we explored emergent model properties in the baseline scenario with fixed resource and temperature conditions, to check that model parameters reproduced reasonable system behaviour. Across the 29 accepted parameter combinations, we observed a wide range of equilibrium biomasses, indicating that these parameters capture a range of alternative system states (Fig E in S1 Text); the modelled equilibrium biomasses were close to the observed interannual variability in species biomasses for the 1990s and 2000s. The emergent resilience to fishing, assessed as fishing mortality at maximum yield in equilibrium conditions, was within the expected range, based on species life-history characteristics (Table F in S1 Text). Emergent diets in model groups reproduced expected ontogenetic shifts, where all species were initially feeding on plankton, with benthivores and herbivores switching to their respective resources, and predators moving from plankton to some benthos and to fish (Fig B in S1 Text). Some benthivore species had small fish in their adult diets, which was consistent with empirical observations [62] and references therein. Emergent relative abundance of benthic resource was close to the available data on average abundance and biomass of benthic invertebrates across the east coast of Tasmania (Fig E in S1 Text).

### Climate change–driven changes in biomasses and yields are strongly determined by food resources

When new resource levels or physiological temperature impacts were applied to the equilibrium biomasses of the baseline scenario, most species settled into new stable or oscillating equilibrium within 20 to 50 years. This suggested that species parameters (mostly the combination of $R_{max}$ and $\varepsilon$) used in the simulations could replicate dynamic species responses and density dependence adjustments to the community. Only 3 model groups were highly sensitive to the explored changes in resource with predicted biomass declining to extremely low levels (extinction) under some parameter combinations. These were the planktivorous serranid *Caesioperca rasor* and urchins, for scenarios where plankton resource carrying capacity ($\kappa$) decreased or slope ($\lambda$) increased, and the wrasse *Pictilabrus laticlavius*, for scenarios with benthic resource changes (Fig 3 and Fig G in S1 Text). Therefore, for these species, uncertainty ranges in biomass and yield responses were very broad. The impact of increasing resource carrying capacity or steeper resource slopes had similar effects on the model group biomasses. This is because steepening resource slopes decreased resource abundances at largest sizes ($>1$ g), but this was offset by complementary increases at smaller sizes ($<1$ g). Below, we only discuss results from changes in resource abundance, whereas impacts of changing resource slope are discussed and shown in Fig G in S1 Text.

In general, physiological responses to warming resulted in predicted changes in species' biomass or yields that were smaller than those from changes in resource abundance (Fig 3). Physiological changes either decreased or did not affect biomasses much, and, more importantly, the biomass response varied across species and trophic groups, despite assuming identical temperature sensitivity parameters for all of them (i.e., increased search, maximum intake and metabolism rates, higher background and senescence mortality at higher temperatures). This indicates that changes in biomasses and yields caused by physiological response to warming were also strongly affected by species interactions, which modified warming-driven intake, metabolism, and mortality impacts.

Changes in plankton resource abundance had similar qualitative effects on biomasses and yields of most species, where biomasses of most species increased by 5% to 10% when plankton

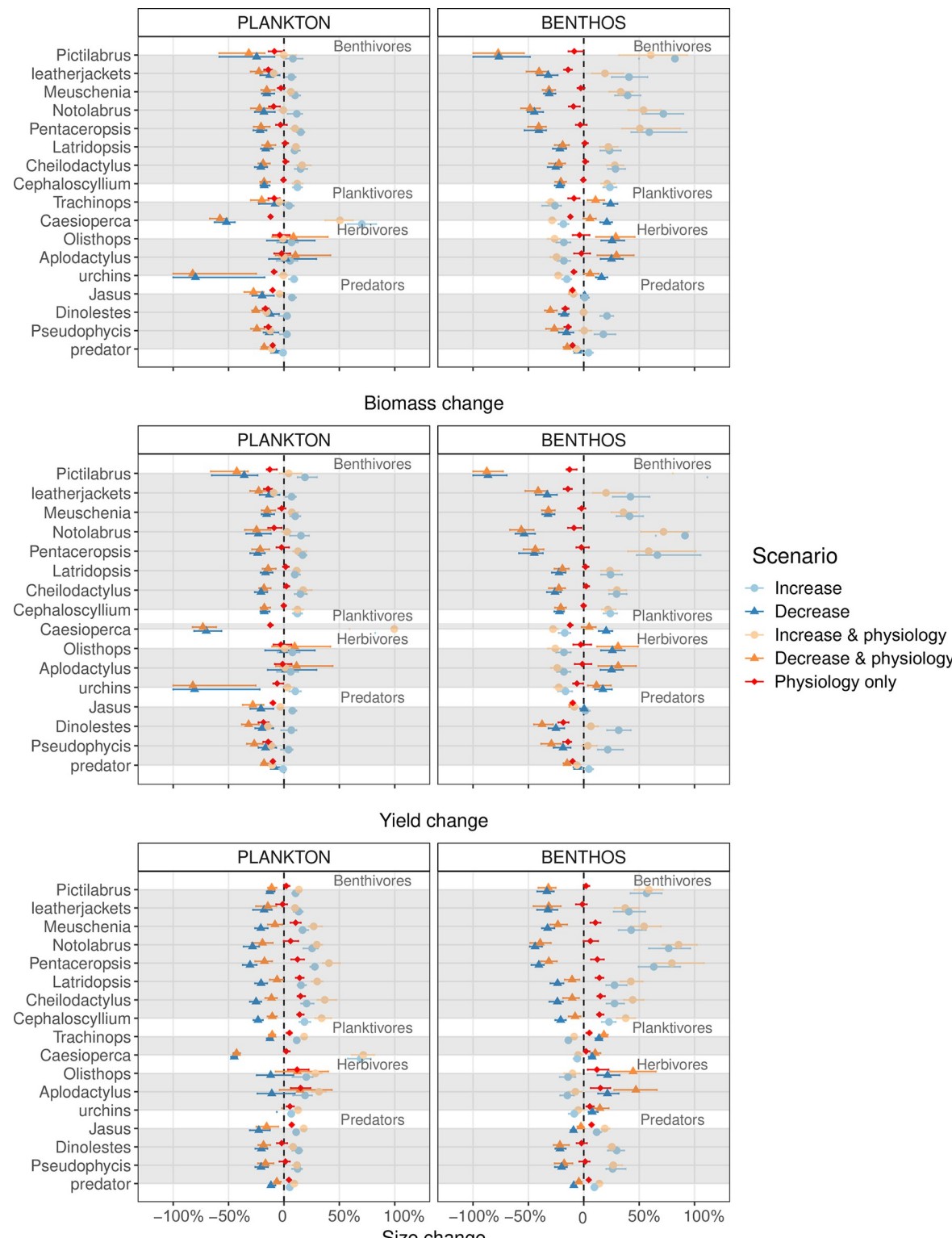

**Fig 3. Relative changes (in %) in biomasses, yields, and mean body sizes (individuals above 2 cm length) in model groups across the alternative model scenarios with increasing or decreasing resource (plankton or benthos) abundance.** Variation across scenarios in all 29 parameter combinations is shown with horizontal bars, depicting 5th and 95th percentiles. Yields of *Pictilabrus* in scenarios with increased benthos abundance where ca. 1.3 (with warming) and 1.8 (no warming) times higher and are not shown here due to scale. Same applies for yields of *Caesioperca* in scenarios with more plankton.

$\kappa$ increased by 30% and decreased by 10% to 20% when $\kappa$ decreased by 30% (with high variation in some species; see light and dark blue symbols in Fig 3). In contrast, changes in benthic resource abundance had opposing effects—biomass and yield of all benthivores changed in the same direction as benthos abundance, whereas the opposite was true for planktivores and herbivores. For predatory species, higher abundance of benthos increased biomass and yields of *Dinolestes lewini* and *Pseudophycis palmata* by ca. 20% but had no effects on lobsters or the general predator.

These contrasting responses across trophic groups were confirmed by the mixed effect ANOVAs (Fig 4 and Table H in S1 Text), where species within a trophic group were treated as random effects, and separate ANOVAs were performed for biomass, yield, and size responses to plankton or benthos changes interacting with physiological change. The strongest response

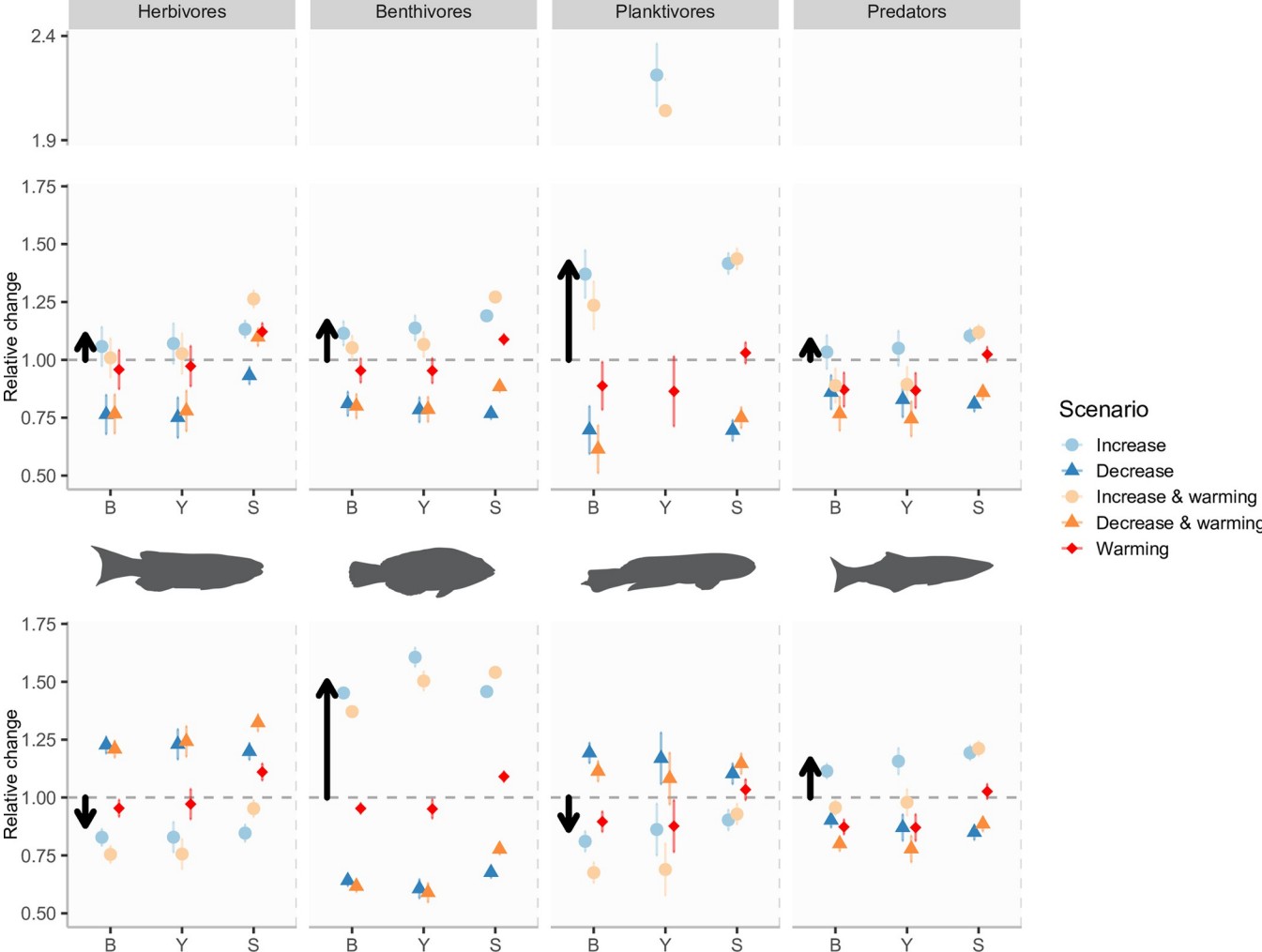

**Fig 4. Predicted effects on biomass, mean size (for individuals larger than 2 cm), and yield of 4 coastal reef fish trophic groups to changes in pelagic and benthic resource abundance and interaction with temperature, under low fishing scenarios, with results of mixed-effect ANOVA.** Error bars show 95% confidence intervals from ANOVA analyses. Black arrows emphasise the predicted effects of increased plankton or benthos abundance on biomass and mean size responses across trophic groups (same as light blue circles) and highlight their contrasting responses to changes in plankton versus benthos. Planktivore yield decreases in reduced plankton scenarios was very strong (ca. 0.25, below Y axis minimum limit shown) and is not shown for clarity. For each parameter combination, all responses are scaled to values in baseline simulations (no resource change, no warming). Note that, for planktivores, biomass and size change reflects effects in 2 species, whereas yield change only shows responses in *Caesioperca* (as *Trachinops* is not fished).

to changes in plankton was, as expected, seen for planktivore biomass and yields, whereas benthivores showed the strongest response to changes in benthic resource. An overall negative response of physiological change was observed in biomass and yields of all 4 trophic groups, but the impact was strongest in predators (red symbols in Fig 4). When physiological responses to temperature were applied together with changes in resource levels, physiology usually had a slightly negative impact on species' biomass, especially for scenarios where resource levels increased (light orange symbols in Figs 3 and 4). This impact varied quantitatively across species and was consistent with their physiological responses to temperature alone; i.e., species that had strongest biomass changes due to change in physiology alone (e.g., *D. lewini*, *P. palmata*) were also more negatively affected by temperature, when physiological responses were combined with the resource change. At a trophic group level, any biomass and yield increase in predators due to higher benthos abundance were completely negated when increasing benthos levels were combined with temperature-driven increases in physiological rates (bottom right, Fig 4). When the general predator species was removed from the model (to explore model sensitivity to the assumptions about its abundance), the overall results about the changes in biomasses, yields, and mean sizes across the simulated scenarios remained very similar (S1 Text section 2.4).

## Faster physiological rates increased mean body sizes of most species, but resource change had larger quantitative effect

As with biomass, the mean intraspecific body size change was considerably more influenced by resource changes than physiological response to warming, and the relative magnitude of mean size change across species (about 10% to 30% across all species) was similar to that of biomass changes (Fig 3). The largest changes in mean size (20% to 60% increase) were seen for benthivorous species under scenarios with increased benthos abundance. Unlike for biomass and yields, however, temperature-driven increases in physiological rates had a positive impact on mean size of most species, regardless of whether physiological rates increased independently of, or in combination with, the resource change. The increase in mean body size caused by physiological responses alone size was strongest in herbivores and benthivores (ca. 10%; red and orange symbols in Fig 4). Strong positive effects of warming on mean body size were likely caused by higher food intake rates at higher temperatures, allowing for faster growth in early life.

Across trophic groups, plankton and benthos changes had similar qualitative effects on species mean sizes as it had for biomass and yields. Namely, when plankton abundance increased, species' mean sizes also increased (or remained similar). In contrast, when benthos abundance increased, only benthivorous and predatory species became on average larger, whereas species in the planktivore and herbivore groups became slightly smaller (Figs 3 and 4).

## Discussion

### Changes in benthic resource abundance have major and contrasting impacts on different trophic groups of coastal fishes

By introducing independent size-structured plankton and benthic production pathways, and exploring how relative changes in their abundance affect coastal fish communities, our study showed that biomasses and mean sizes of different trophic groups can respond to benthic resource changes in qualitatively different ways. Increased abundance of benthic organisms led to higher biomasses, yields, and mean sizes of benthivorous and predatory fishes (and lobsters), but the reverse was true for herbivores and planktivores. In contrast, changes in

plankton abundance induced similar responses across all trophic groups, such that decreasing plankton abundance affected all species negatively.

To date, marine ecosystem climate change studies and regional and global ocean models have been strongly focused on understanding changes in plankton primary production, which drives fish production in pelagic and shelf ecosystems and supports huge fisheries [75]. Our findings are generally consistent with these studies showing that decreasing plankton primary productivity leads to an overall decrease in fish biomass, although the magnitude of the change depends on the system [8,16,17]. Such overarching and consistent influence of plankton abundance on all fish is not surprising, given that most fish species (as larvae or juveniles) initially feed on plankton.

A key novel finding of our study is the critical importance of the benthic production pathway in shallow coastal rocky reef ecosystems, with implications for all shallow marine and freshwater ecosystems that can support a diversity of benthic primary producers (algae) and invertebrates that depend on them. Abundance and size structure of benthic organisms in shallow water ecosystems are key determinants of ecosystem dynamics and fish community structure [76] yet remain largely unmonitored and vastly underrepresented in ecosystem and community models. For example, the benthic production pathway in lake ecosystems is believed to provide at least 50% of whole lake productivity, yet 91% of 193 lake food web studies reviewed by [77] considered only plankton productivity. Remote observation data can now provide high resolution and quality estimates of plankton abundance at local and global scales, yet spatial and temporal variation in benthic energy pathways remains poorly understood in shallow marine systems, and even less is known about how these pathways are responding to the changing climate.

Monitoring and assessing benthic production changes and their impacts on shallow water ecosystems is particularly important for climate change predictions, because changes in benthic community structure could be even larger than in plankton production. In addition to changing sea temperatures and nutrient profiles that affect open ocean habitats, the coastal seas, lagoons, and freshwater lakes will also experience productivity changes resulting from climate-driven precipitation changes on land, altered land-use, and urbanisation. Serious widespread warming-driven declines in temperate benthic communities have recently been highlighted by a comprehensive study of changes in reef species abundances around Australia over the last decade [78]. Temperate benthic invertebrates were among the most severely affected components of reef communities, with more than 30% of species experiencing rapid population declines and an average of 18% decline in abundance among all cool temperate invertebrate species studied.

Our model suggests that declining benthic productivity can lead to increased herbivore and planktivore biomasses and yields, trends that have been observed along Tasmanian coasts in the reef monitoring data (see Fig G in S1 Text showing observed changes overlaid over the model predictions of resource slope changes, noting that the simulations used in this study did not represent real plankton and benthos changes that have likely occurred in the Tasmanian rocky reefs). Although herbivore increases were almost certainly driven by other processes, such as water temperatures exceeding physiological thresholds and associated species redistributions, the opposing responses of different trophic groups to changes in benthos resource identified in our study emphasise the importance of species interactions and possible feedback loops [79]. Our results thus imply that changes in benthos could lead to a rapid reorganisation of fish communities and food webs. It is therefore imperative that more effort is placed on collecting monitoring data on the benthic production pathway and that we develop general and consistent methods to integrate this pathway into regional and global ecosystem models.

The importance of integrating benthic organisms into community size spectra and ecosystem models that aim to explore climate change effects is well recognised [80,81], although we still lack a consistent approach to represent this benthic community diversity in necessarily simplified models. Benthic organisms are often included in biogeochemical models that explore nutrient flows and eutrophication effects [82], but these models do not explore impacts and interactions with fish communities. Applications of mass balance (e.g., Ecosim/Ecopath) or end-to-end ecosystem models (e.g., Atlantis, OSMOSE) sometimes also include key benthos groups such as worms, bivalves, shrimps, and similar [48,64,83]. However, in such studies, benthic organisms either are modelled as unstructured biomass pools or require detailed parameterisation to represent age-based growth dynamics and still do not represent the entire benthic production pathway from primary producers to invertebrates. Modelling the diversity of benthic functional groups is a daunting and possibly unnecessary task, because despite the large variation in taxonomic groups and the overall abundance, the size structure of the benthic organisms can remain similar across large temporal and spatial scales [53,55,84]. This means that representing the benthic production pathway as a size-structured resource, as is introduced in this study, might provide an effective and general approach for both coastal and freshwater ecosystems.

Preserving size structure of the benthic community is important both because it allows a simplified representation of diverse groups and because benthic organisms of different body size can respond to climate change in opposing ways [84]. In our study, steepening of benthic resource size slopes and a resultant shift of resource abundance towards smaller organisms had an overall positive impact on benthivore fish biomass (Fig G in S1 Text). This is because most food limitation occurred at small fish body sizes, and large predator–prey mass ratios of benthivorous and planktivorous fishes [85] means that most fishes were feeding on smallest benthic size groups. Of course, the sensitivity of fish community to changes in resource size spectrum slope, and, therefore, relative abundances at size, will depend on the parameters defining size-based feeding, and the sensitivity of these parameter should be explored in further studies. Future work could also explore whether multiple size-structured resources, as modelled here, could be used to represent habitat complexity, which could interact with fish recruitment or fish predation vulnerability. In coral reef ecosystems, adding size-based predation refuges improved the fit of modelled fish size spectra to empirical observations [32]. For temperate rocky reefs, habitat complexity is also provided by large kelp stands, which we approximated with the macroalgal size-structured resource. We did not model interactions between different size spectra (e.g., macroalgal spectrum affecting regeneration rate of benthic spectrum) or impacts of size-structured resource on fish recruitment or predation vulnerability, and this would be an important future exploration.

Of course, conclusions from our study are highly dependent on model assumptions and simplifications of a complex system. While we show that resource changes have greater impacts than temperature changes, clearly, this will depend on the relative magnitude of change assumed in the model. In our simulations, the impacts of both productivity and temperature changes were large (30% change in abundance) but generally consistent with recent findings on benthic invertebrate declines in Tasmania [78,86] and within a realistic range for plankton (see S1 Text section 1.7). More specific scenarios were not possible due to high uncertainty in the expected global warming–driven productivity and size structure changes, especially for the benthos [84,87]. Further, we only explored changes in pelagic or benthos resource separately, when in reality, abundance and size structure of both plankton and benthic resources have most certainly changed simultaneously. It was not feasible to address all possible resource and temperature change scenarios here, but we provide an online ShinyR

tool (https://fishsize.shinyapps.io/BenthicSizeSpectrum) that allows simulation of a wide range of alternative scenarios to hopefully encourage further studies.

## Physiological responses to temperature lead to lower biomass and larger mean sizes in individual species, but community-level impacts are relatively small

Compared to changes in plankton or benthos abundance and size structure, the predicted physiological consequences of warming seas on biomass and mean size of fishes and large invertebrates were relatively small. Notably, even though our model assumed the same temperature response parameters in all modelled species, the relative magnitude of physiology-driven impacts varied across species and trophic groups. These results show that in natural communities, observed body size and biomass changes due to warming are likely strongly shaped by species interactions and therefore hard to predict a priori. In our study, physiological changes, both when analysed separately and in combination with resource change, had large impacts on predator biomass and yields but very little on herbivore and benthivore biomass (Fig 4). For predators, such as *Dinolestes lewini* or the commercially valuable and ecologically important rock lobster, the negative physiological effect of increasing temperature negated any positive impacts on their biomass from increasing benthos abundance. This appears to be a food limitation response, in which these predators were unable to compensate increased metabolic needs with higher food intake. Our finding is in line with other recent modelling studies [88] and empirical observations demonstrating that fish responses to warming are mediated by food availability.

To keep the number of scenarios tractable and to explore the role of species interactions in modifying physiological or resource-driven changes, we assumed that all species and all vital rates (search rate, maximum food intake, metabolism, background, and senescence mortality) scaled with temperature in the same way. This is consistent with the metabolic theory of ecology [89] and still remains one of the most common approaches used in a understanding ecosystem responses to warming (e.g., see the summary of the diverse models used in Fisheries and Marine Ecosystem Model Intercomparison Project FishMIP [22]). In reality, temperature responses will certainly differ across rates, species, ontogeny, populations, and through time due to acclimation and adaptation [90,91]. Yet, much uncertainty remains about the magnitude and even direction of these differences, let alone the most robust way to model them. Some models have assumed that metabolism scales with temperature exponentially, but intake has a domed shaped and species-specific response [92]. Others have assumed that metabolism costs become relatively more expensive in larger animals and have faster scaling with temperature compared to intake [93]. Alternatively, the uncertainty of metabolism and intake response to temperature could be treated as part of model parameter uncertainty, such as found by [88], who sampled a range of activation energies for these 2 rates (still assuming that both rates scale exponentially). Such temperature response uncertainty evaluation was not possible in our study, given that we already included extensive uncertainty evaluation for reproduction, intake, and interaction parameters among 37 species. Finally, there remains a big uncertainty about adequately representing acclimation and adaptation to temperature changes. On one hand, there is good evidence that given sufficient intergenerational acclimation time, fish or plankton in experimental conditions can maintain similar metabolic rates across a range of temperatures [94,95]. On the other hand, physiological rates do vary across temperatures [89], which suggests that they might be driven by the overall community reorganisation to a faster pace of life (see discussion in [95]). Given that such reorganisation is expected in natural systems, it might be more appropriate to simulate climate change impacts without the acclimation

observed in laboratory conditions. More field-based temporal studies of metabolic rates in warming hotspots are needed to resolve this question.

Nevertheless, even with our simple physiological response assumption, we found that the model-predicted magnitude of mean body size changes due to physiology alone (red symbols in Fig 3) varied substantially across species (0% to 15% change). Moreover, when model uncertainty was taken into account, no species showed a consistent decrease in mean body size. This may seem to contradict many experimental and modelling studies, which predict that warming waters will lead to smaller fish [96–98] and that this "shrinking" is due to temperature-driven physiological change [89,99,100]. Yet, our model findings of an overall increase in mean size due to physiological response do not necessarily contradict the "shrinking" fish paradigm because this paradigm refers to temperature impacts on maturation and maximum body size rather than population means [101].

Warming impacts on mean species sizes will be complex because they will depend on both physiology and growth and also changes in mortality. In fact, even changes in benthos abundance produced qualitatively different mean body responses in different species and trophic groups, where some species became, on average, larger and others smaller. Such differential response is consistent with findings from a large-scale empirical study of 335 fish species around the Australian continent, which showed that warming across space or time led to mean species body sizes decreasing in 55% of fish species and increasing in 45% [74]. If we plot the mean body size changes of Tasmanian coastal species reported in [74] against the mean size changes predicted in our model simulations (black dots in Fig G in S1 Text), we find that empirically observed mean body size changes were most consistent with predictions from scenarios with 30% decreases in plankton or benthos abundance (dark orange and blue symbols in Fig 3C). However, we emphasise that real changes in Tasmanian reefs have certainly been more complex than our scenarios, and empirical changes are only shown to illustrate the magnitude of fish community changes occurring in this coastal ecosystem. Regardless, our study shows that complex and diverse changes in fish communities can emerge even under simple assumptions and that species interactions must be considered when aiming to predict how marine ecosystem are likely to change over the next decades.

While simple physiological response assumptions were a useful first step to explore interactions across resource, physiology, and fish community processes, future studies should explore more complex scenarios of species response to warming. The first important improvement would be to assess the interaction of temperature-driven changes in vital rates and life-history trade-offs such as relative energy allocation to growth and reproduction [66,102,103]. For example, Audzijonyte and colleagues [104] used an individual growth model with life-history optimisation and showed that accounting for life-history changes leads to more realistic emergent growth and maturation responses. To our knowledge, however, none of the commonly applied multispecies models (e.g., the diversity of models in the FishMIP) incorporate potential temperature-driven changes in maturation or reproduction, although such changes are expected and well documented [101,105]. Some degree of growth-reproduction trade-off does occur in size-based models and *mizer*, because growth and maturation age are emergent model properties rather than being set [106]. Yet, the current trade-offs appear to be insufficient to fully account for possible reproduction costs and life-history optimisation processes [107]. The second important improvement would be to consider different temperature responses across species and physiological rates and, especially, size dependency of these responses [60]. Some of this exploration will hopefully be facilitated by the *mizer* add-on package *therMizer* (https://github.com/sizespectrum/ therMizer) that enables simulating domed shaped intake responses, as used in [10]. Size dependency of vital rate acceleration can also be

relatively straightforward to incorporate using the size dependency exponent added to the Arrhenius equation [104].

## Parameter uncertainty evaluation and ecological realism

Parameter uncertainty evaluation in complex models remains one of the major challenges in ecological modelling, and the uncertainty evaluation approach used in this study presents 2 important improvements. The first one is to add more ecological realism to complex models assessing their performance and skill against a broad range of ecological criteria. Traditionally, calibration of size-based multispecies models focuses on species recruitment and resource abundance parameters, evaluated against observations of species biomasses or time series of fisheries catches when fishing mortality in the model is the same as mortalities from stock assessments [34,43,63]. Such calibrations can be done using model optimisation procedures where observational error between observed and simulated catches is minimised using least square or other mathematical criteria. While this is an important step, the parameter optimisation approach only delivers one set of "optimal" parameters, but given the complexity of the parameter space, it is almost certain that multiple parameter combinations could match the limited observations against which the optimisation was conducted. This means that we should be exploring model performance against a much larger range of observations, even if these "observations" may not be formally quantified.

Multispecies models have many emergent ecological and biological properties, such as fish growth rates, diets, and interactions, and we often have reasonable ecological knowledge about the possible ranges of these properties. This ecological knowledge is hard to fit into an optimisation function, which is why it is often ignored or used only informally in the model exploration stages. Incorporating a broad range of ecological knowledge in parameter selection would increase the ecological realism of models. For example, species biomasses or catches may fit the observations perfectly, yet species diets, growth, and interactions may be unrealistic. This is why an increasing number of modellers call for more ecological realism of models to ensure that "correct outcomes are predicted for correct reasons" [69]. We have attempted to add such realism, by assessing model performance with over 2 million parameter combinations against a list of ecological criteria, derived from expert and general ecological knowledge. Many of these parameter combinations fitted expectations about species biomasses but completely failed other ecological criteria (e.g., diets), suggesting that model realism would have been considerably worse if additional ecological criteria were not incorporated.

By adding parameter uncertainty into our study, we are explicitly addressing uncertainty in community responses to alternative climate change impacts, which could be formalised in an ANOVA-type analysis design. The parameter uncertainty evaluation in this study is different from model sensitivity analyses where parameters are drawn randomly from a specified distribution around the initial optimised parameter value and are all considered equally possible, e.g., as in [108] or [34]. Our analyses are more similar to the Bayesian uncertainty evaluation of stock recruitment and plankton abundance parameters in the North Sea size-based model [43], except that we applied a simpler parameter evaluation framework that did not require specifying a likelihood function, allowing us to incorporate multiple ecological criteria (not used in [43]). Naturally, many aspects of parameter uncertainty evaluation could be improved. While the current study presents one of the most exhaustive parameter uncertainty evaluations across a range of different traits (recruitment, food intake rates, and species interactions), we did not explore the uncertainty in species-specific physiological parameters and how these might affect emergent species body sizes and abundance. We evaluated model outcomes against a set of ecological criteria, effectively assuming a knife-edge likelihood function (e.g.,

proportion of one species in a diet of another species cannot be smaller than a certain fixed value). This method could be improved by using a smoother function. Finally, machine learning tools should also be used to find parameter patterns from multiple model runs against multiple ecological criteria and, hopefully, to incorporate other aspects of model uncertainty in addition to parameter values.

## Future directions

Adequate prediction of coastal community responses to warming requires an understanding of interactive resource and physiological changes in multispecies contexts. Yet, marine ecosystem responses to climate change also involve other factors with potentially large impacts. The main one relates to the possible impacts of species redistributions [109], where newly arriving species (e.g., sea urchins in kelp forest ecosystems) might cause major ecosystem shifts. Redistribution of herbivorous fish species into temperate reefs is already driving large changes in macroalgal abundance and community composition [110]. The second major one is human impacts, through either increased fishing or increased protection. An increasing number of studies shows that protecting large fishes, and especially predators, can help mitigate population and ecosystem impacts of warming [111,112]. Our study did not explore how different fishing scenarios might interact with global warming, but such analysis would be very important. Finally, the current framework also allows assessment of climate change impacts on broader community attributes and indicators used in ecosystem management and ecology. These include such characteristics as resilience and community connectedness, and ecological and fisheries indicators based on body size distributions in fish communities, overall community level size structure and maximum sustainable yield, or ecosystem-level energy transfer efficiency. Our study provides a pathway that will hopefully encourage and facilitate further investigation.

## Supporting information

**S1 Text. Supplementary methods and results.**
(DOCX)

## Acknowledgments

We thank Gretta Pecl and Jonathan Reum for suggestions for this study, Freddie Heather for help with the ShinyR tool, Amy Coghlan for model illustration (Fig 2), Lara Beckmann for help with Fig 4, and many volunteers who helped with RLS surveys. The ecological data used for this study are managed through, and were sourced from, Australia's Integrated Marine Observing System (IMOS)—IMOS is enabled by the National Collaborative Research Infrastructure Strategy (NCRIS).

## Author Contributions

**Conceptualization:** Asta Audzijonyte, Gustav W. Delius, Julia L. Blanchard.

**Data curation:** Asta Audzijonyte, Rick D. Stuart-Smith, Graham J. Edgar, Neville S. Barrett.

**Formal analysis:** Asta Audzijonyte, Gustav W. Delius, Camilla Novaglio, Julia L. Blanchard.

**Funding acquisition:** Asta Audzijonyte, Julia L. Blanchard.

**Investigation:** Asta Audzijonyte, Gustav W. Delius, Julia L. Blanchard.

**Methodology:** Asta Audzijonyte, Gustav W. Delius, Julia L. Blanchard.

**Project administration:** Asta Audzijonyte, Julia L. Blanchard.

**Resources:** Asta Audzijonyte, Gustav W. Delius, Rick D. Stuart-Smith, Graham J. Edgar, Neville S. Barrett, Julia L. Blanchard.

**Software:** Asta Audzijonyte, Gustav W. Delius, Julia L. Blanchard.

**Supervision:** Asta Audzijonyte, Julia L. Blanchard.

**Validation:** Gustav W. Delius, Camilla Novaglio.

**Visualization:** Asta Audzijonyte, Gustav W. Delius, Camilla Novaglio, Julia L. Blanchard.

**Writing – original draft:** Asta Audzijonyte, Rick D. Stuart-Smith, Julia L. Blanchard.

**Writing – review & editing:** Asta Audzijonyte, Gustav W. Delius, Rick D. Stuart-Smith, Camilla Novaglio, Graham J. Edgar, Neville S. Barrett, Julia L. Blanchard.

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
