## [Editor Report · Decision Letter 0]

15 Sep 2022

Dear Dr Delius, 

Thank you for submitting your manuscript entitled "Changes in benthic and pelagic production interact with warming to drive responses to climate change in a temperate coastal ecosystem" for consideration as a Research Article by PLOS Biology.

Your manuscript has now been evaluated by the PLOS Biology editorial staff, as well as by an academic editor with relevant expertise, and I'm writing to let you know that we would like to send your submission out for external peer review.

Once your full submission is complete, your paper will undergo a series of checks in preparation for peer review. After your manuscript has passed the checks it will be sent out for review. To provide the metadata for your submission, please Login to Editorial Manager (https://www.editorialmanager.com/pbiology) within two working days, i.e. by Sep 19 2022 11:59PM.

Kind regards,

Roli Roberts

Roland Roberts, PhD

Senior Editor

PLOS Biology

rroberts@plos.org

---

## [Decision Letter · Decision Letter 1]

7 Nov 2022

Dear Dr Delius,

Thank you for your patience while your manuscript "Changes in benthic and pelagic production interact with warming to drive responses to climate change in a temperate coastal ecosystem" was peer-reviewed by PLOS Biology. Your work was assessed and discussed by the PLOS Biology editorial team, an Academic Editor with relevant expertise, and by two independent reviewers. Based on the reviews, which you will find at the end of this email, I regret that we will not be pursuing your manuscript for publication at PLOS Biology.

You will see that the two reviewers have quite divergent opinions of the manuscript. We also invited the reviewers to cross-comment on each other's assessment, and then discussed both the reviews and these cross-comments at length with the Academic Editor. While we recognise that both reviewers appreciate what you are attempting to do, unfortunately our conclusion was that the concerns raised by reviewer #2 were very persuasive, and that on these grounds we should decline to consider your manuscript further.

I am sorry that we cannot be more positive on this occasion and hope the reviewer reports will help you in preparing your manuscript for submission elsewhere.

While we cannot consider your manuscript further for publication in PLOS Biology, we would suggest transferring your manuscript, with reviews, to PLOS ONE instead (http://journals.plos.org/plosone/). 

PLOS ONE is a peer-reviewed journal that accepts original research that contributes to the base of academic knowledge. The review process at PLOS ONE focuses on scientific validity, strong methodology and high ethical standards, and the journal's inclusive scope and broad reach means that research published in PLOS ONE will be read, cited and used by researchers across many disciplines. In this case, the PLOS ONE Academic Editors will also take the feedback received from the reviewers at PLOS Biology into account when reaching a decision, which should increase the efficiency of the review process. Please note that the journals are editorially independent and we therefore cannot guarantee the outcome if you choose to pursue a transfer. 

If you would like to submit your work to PLOS ONE, please click the following link:

<DeepLinkData><DeepLinkTypeID>27</DeepLinkTypeID><peopleID>1563451</peopleID><userSecurityID>255ae21e-2991-45e3-bab6-442d6a8e1802</userSecurityID><documentID>50869</documentID><revision>1</revision><manuscriptNumber>PBIOLOGY-D-22-01984</manuscriptNumber><docSecurityID>78cdb72b-e0c4-4f2e-b1e8-abed4bd0c981</docSecurityID></DeepLinkData>

If you do NOT wish to submit your work to PLOS ONE, please click this link to decline: 

<DeepLinkData><DeepLinkTypeID>28</DeepLinkTypeID><peopleID>1563451</peopleID><userSecurityID>255ae21e-2991-45e3-bab6-442d6a8e1802</userSecurityID><documentID>50869</documentID><revision>1</revision><manuscriptNumber>PBIOLOGY-D-22-01984</manuscriptNumber><docSecurityID>78cdb72b-e0c4-4f2e-b1e8-abed4bd0c981</docSecurityID></DeepLinkData>

Should you choose to transfer your submission to PLOS ONE, you will receive a confirmation email within 24-48 hours of accepting the transfer. Your submission details and manuscript files will be transferred automatically; however, because all PLOS journals vary in submission requirements, once in the PLOS ONE Editorial Manager site, you will be asked to provide additional information before you can finalize your new submission to PLOS ONE. If you have any questions, please feel free to contact the journal at plosone@plos.org.

I hope you understand the reasons for this decision and that the option of publishing your work in PLOS ONE might be useful. Thank you for your support of PLOS and of open-access publishing.

Sincerely,

Roli Roberts

Roland Roberts, PhD

Senior Editor

PLOS Biology

rroberts@plos.org

REVIEWERS' COMMENTS:

Reviewer #1:

This paper used a modeling approach to couple changes in planktonic and benthic production and warming effects on physiological processes of individuals and explored how these two things might, independently and jointly, alter food web dynamics in an Australian costal community. In doing so, the authors implemented well established size-spectrum models by explicitly separating the planktonic and benthic resource pathways and introduced a novel way of calibrating the model without using observed biomass or catch data. The implementation of the two resource pathways is also available as an R package. The main finding of this paper was that changes in resource dynamics generally had much stronger effects on species yields and biomasses than did those due to warming-related effects on physiology. Planktivores and benthivores were impacted more by changes in planktonic and benthic resources, respectively, and these impacts cascaded through the food web. 

I liked the factorial design for assessing parameter uncertainty; using it to find reasonable potential results without the need of catch data was very clever. This is very significant because it enables people to construct credible multispecies size spectrum models for data-poor communities or for future climates, which was very challenging, if not impossible, before. The authors' effort to pursue ecological realism is detailed and impressive. Overall, I found this paper to be very illuminating. 

As one of the evaluation criteria, the only place that this paper could benefit from additional data is a broader consideration of warming effects. This paper took into account how warming would impact species' metabolic rate, search rate during foraging, intake rate, and predation-unrelated mortality. Even though none of these impacts is trivial, there are other warming consequences that could be at least equally important, such as changes in key life-history traits and shifts in species distribution, both of which can influence the modeling outcomes. I am not suggesting a total overhaul of the analyses, but it would be informative if there could be more discussions on whether (and how) this might have been behind some of the notable discrepancies between predicted and observed species biomasses (the authors did touch on these aspects in the Discussion, but only briefly). 

Lastly, I also wanted to highlight that this paper had one of the most thorough and useful Supplementary Materials I've ever read. It would be even nicer if the authors could refer to specific sections of the SM in the main text. 

I also listed several more minor comments below, most of which relate to methodological clarity. I hope my comments were helpful. I thank the authors for producing such an enjoyable and informative paper. 

Minor comments

Line 159-162, Table 1 and Table S3. I like the approach of including a generalized large predator to account for their often-underestimated abundance, but how did the authors decide on the values for maximum weight, weight at maturity, and fishing mortality for this group? A maximum weight of 5 kg seems rather small for representing large predators. The choice of these values likely won't impact the results dramatically, but some explanations/justifications may be necessary here. 

Figure S3 was supposed to show species' responses to fishing based on their size, but that was not what was shown. It also looked like the references to the supplementary figures in the main text were off by one after Fig S2.

Table S5. The authors mentioned in the Supplement that the availability of the three resources to each species were given in Table S5, but I didn't see this information there. Maybe the authors accidentally left it out?

Line 402-406. It was not clear to me how the number of 1.8e6 was obtained from exploring two possible values of 37 variables (0.5 x and 2 x the original values). I intuitively thought that there would be a total of 2^37 parameter combinations, but this would produce a much larger number (~1.4e11). How did the authors decide which 1.8e6 combinations to sample? The same goes for the additional 0.4e6 samples (0.8 x and 1.2 x the original values). It might be worth adding one short sentence to explain this better.

Line 417. I like the author's approach of filtering out inappropriate parameter combinations by imposing extra fishing mortality. However, the authors seemed to have made several subjective choices regarding the threshold of responses beyond which parameters would be considered unrealistic (criteria 1-4: 0.33-3 x for fishing lobsters, 0.67 - 1.5x for fishing sea urchins, etc). I am not questioning these choices per se, but it would be helpful if the authors could explain their rationale behind these choices. It would help readers who would like to use a similar approach to decide whether they should use the same thresholds for their systems or, if not, how they should modify them.

Reviewer #2:

This is a dense, parameter-rich exploration of how pelagic and benthic communities might change. The modelling is fairly straightforward, based on previous approaches used by this group of researchers and others. The model makes many predictions, depending on the scenario and inevitably, with so many predictions, some match real world changes. Give the shear number of parameter combinations (~30), it's unclear to me that this is valuable exercise because with so many components, it's near impossible for a model not to make some predictions that reflect reality. 

Yet with all of these parameters, the authors then take the oversimplified approach of assuming that all temperature-dependent processes have the identical temperature dependencies - an old-fashioned approach that has repeatedly been debunked since it was first proposed. To me, that approach is inadequate in this specific work for several reasons. First, we know that different processes have very different temperature dependencies. Second, we know that slight mismatches in temperature dependencies yield very different net outcomes from opposing processes. Finally, as the authors mention, these temperature dependencies themselves change with temperature through both evolution and plasticity. 

All in all, then for a paper that seeks to explore how warming affects function, the actual warming component is massively overly simplified - reduced to uniform rate functions. Meanwhile, the the authors explore such a massive range of scenarios that some congruence with reality is inevitable but not informative. In other words, because data are lacking on the key drivers of warming effects, the authors make strong, outdated assumptions about uniform temperature dependence while exploring so many different input scenarios that the model predictions include (among many others) responses that are similar to observations. To me, this is a good example of where we don't have the data to make realistic useful models to attack this problem but, with enough parameters and scenarios, models can make so many predictions, that they look like they are capturing the process, when they are not. I'm very sorry I cannot be more positive but I don't think this model achieves the ambitious goals that it sets itself because we simply lack the data we need to meaningfully model these processes.

---

## [Editor Report · Decision Letter 2]

18 Jan 2023

Dear Dr Delius,

Thank you for your patience while we considered your Appeal of our previous decision to decline to consider your manuscript "Changes in benthic and pelagic production interact with warming to drive responses to climate change in a temperate coastal ecosystem" further.

As advised in my previous email, we noted that you felt that reviewer #2 was "misunderstanding the goals and extent of our study"; given this reviewer's expertise, we think that this is unlikely (for example, your conclusion that this reviewer is "not necessarily experienced in community and ecosystem modelling studies" is not correct). However, in discussions with the Academic Editor, they pointed out that while reviewer #2 almost certainly *does* fully understand what you are attempting to do, s/he has a particular opinion of your study's relevance to real life scenarios which may or may not be shared by others in the field. We therefore propose to seek further opinions (by further peer review) in order to establish the degree of broader interest in your work.

In light of the reviews, which you will find at the end of this email, we would like to invite you to revise the work to thoroughly address the reviewers' reports; when you re-submit, we will seek input from additional reviewers in order to determine a more representative view of the importance of your work to the field.

Given the extent of revision needed, we cannot make a decision about publication until we have seen the revised manuscript and your response to the reviewers' comments. Your revised manuscript will be sent for further evaluation by all or a subset of the existing reviewers, plus one or two additional ones.

**IMPORTANT - SUBMITTING YOUR REVISION**

*Re-submission Checklist*

*Published Peer Review*

*PLOS Data Policy*

*Blot and Gel Data Policy*

Sincerely,

Roli Roberts

Roland Roberts, PhD

Senior Editor

PLOS Biology

rroberts@plos.org

REVIEWERS' COMMENTS:

Reviewer #1:

This paper used a modeling approach to couple changes in planktonic and benthic production and warming effects on physiological processes of individuals and explored how these two things might, independently and jointly, alter food web dynamics in an Australian costal community. In doing so, the authors implemented well established size-spectrum models by explicitly separating the planktonic and benthic resource pathways and introduced a novel way of calibrating the model without using observed biomass or catch data. The implementation of the two resource pathways is also available as an R package. The main finding of this paper was that changes in resource dynamics generally had much stronger effects on species yields and biomasses than did those due to warming-related effects on physiology. Planktivores and benthivores were impacted more by changes in planktonic and benthic resources, respectively, and these impacts cascaded through the food web.

I liked the factorial design for assessing parameter uncertainty; using it to find reasonable potential results without the need of catch data was very clever. This is very significant because it enables people to construct credible multispecies size spectrum models for data-poor communities or for future climates, which was very challenging, if not impossible, before. The authors' effort to pursue ecological realism is detailed and impressive. Overall, I found this paper to be very illuminating.

As one of the evaluation criteria, the only place that this paper could benefit from additional data is a broader consideration of warming effects. This paper took into account how warming would impact species' metabolic rate, search rate during foraging, intake rate, and predation-unrelated mortality. Even though none of these impacts is trivial, there are other warming consequences that could be at least equally important, such as changes in key life-history traits and shifts in species distribution, both of which can influence the modeling outcomes. I am not suggesting a total overhaul of the analyses, but it would be informative if there could be more discussions on whether (and how) this might have been behind some of the notable discrepancies between predicted and observed species biomasses (the authors did touch on these aspects in the Discussion, but only briefly).

Lastly, I also wanted to highlight that this paper had one of the most thorough and useful Supplementary Materials I've ever read. It would be even nicer if the authors could refer to specific sections of the SM in the main text.

I also listed several more minor comments below, most of which relate to methodological clarity. I hope my comments were helpful. I thank the authors for producing such an enjoyable and informative paper.

Minor comments

Line 159-162, Table 1 and Table S3. I like the approach of including a generalized large predator to account for their often-underestimated abundance, but how did the authors decide on the values for maximum weight, weight at maturity, and fishing mortality for this group? A maximum weight of 5 kg seems rather small for representing large predators. The choice of these values likely won't impact the results dramatically, but some explanations/justifications may be necessary here.

Figure S3 was supposed to show species' responses to fishing based on their size, but that was not what was shown. It also looked like the references to the supplementary figures in the main text were off by one after Fig S2.

Table S5. The authors mentioned in the Supplement that the availability of the three resources to each species were given in Table S5, but I didn't see this information there. Maybe the authors accidentally left it out?

Line 402-406. It was not clear to me how the number of 1.8e6 was obtained from exploring two possible values of 37 variables (0.5 x and 2 x the original values). I intuitively thought that there would be a total of 2^37 parameter combinations, but this would produce a much larger number (~1.4e11). How did the authors decide which 1.8e6 combinations to sample? The same goes for the additional 0.4e6 samples (0.8 x and 1.2 x the original values). It might be worth adding one short sentence to explain this better.

Line 417. I like the author's approach of filtering out inappropriate parameter combinations by imposing extra fishing mortality. However, the authors seemed to have made several subjective choices regarding the threshold of responses beyond which parameters would be considered unrealistic (criteria 1-4: 0.33-3 x for fishing lobsters, 0.67 - 1.5x for fishing sea urchins, etc). I am not questioning these choices per se, but it would be helpful if the authors could explain their rationale behind these choices. It would help readers who would like to use a similar approach to decide whether they should use the same thresholds for their systems or, if not, how they should modify them.

Reviewer #2:

This is a dense, parameter-rich exploration of how pelagic and benthic communities might change. The modelling is fairly straightforward, based on previous approaches used by this group of researchers and others. The model makes many predictions, depending on the scenario and inevitably, with so many predictions, some match real world changes. Give the shear number of parameter combinations (~30), it's unclear to me that this is valuable exercise because with so many components, it's near impossible for a model not to make some predictions that reflect reality.

Yet with all of these parameters, the authors then take the oversimplified approach of assuming that all temperature-dependent processes have the identical temperature dependencies - an old-fashioned approach that has repeatedly been debunked since it was first proposed. To me, that approach is inadequate in this specific work for several reasons. First, we know that different processes have very different temperature dependencies. Second, we know that slight mismatches in temperature dependencies yield very different net outcomes from opposing processes. Finally, as the authors mention, these temperature dependencies themselves change with temperature through both evolution and plasticity.

All in all, then for a paper that seeks to explore how warming affects function, the actual warming component is massively overly simplified - reduced to uniform rate functions. Meanwhile, the the authors explore such a massive range of scenarios that some congruence with reality is inevitable but not informative. In other words, because data are lacking on the key drivers of warming effects, the authors make strong, outdated assumptions about uniform temperature dependence while exploring so many different input scenarios that the model predictions include (among many others) responses that are similar to observations. To me, this is a good example of where we don't have the data to make realistic useful models to attack this problem but, with enough parameters and scenarios, models can make so many predictions, that they look like they are capturing the process, when they are not. I'm very sorry I cannot be more positive but I don't think this model achieves the ambitious goals that it sets itself because we simply lack the data we need to meaningfully model these processes.

---

## [Decision Letter · Decision Letter 3]

13 Jul 2023

Dear Dr Delius,

Thank you for your patience while we considered your revised manuscript "A new size-based model highlights the importance of benthic production changes for understanding climate change impacts on a temperate coastal ecosystem" for consideration as a Research Article at PLOS Biology. Your revised study has now been evaluated by the PLOS Biology editors, the Academic Editor and one of the original reviewers. In addition, we recruited a new reviewer (reviewer #3), whom we invited to comment on the divergent opinions of reviewers #1 and #2, and to provide an independent assessment of the revised manuscript. Please accept my apologies for the extreme delay incurred; I'd like to stress that this was down to our own difficulties in recruiting the new reviewer, and is not the fault of either the reviewers or the Academic Editor.

You'll see that reviewer #1 is now satisfied. Reviewer #2 advised us that they did not think it fair that they should re-review the manuscript, and that we should seek independent advice. Reviewer #3, who has a similar expertise to reviewer #2, is positive about your study, but has a few minor concerns and requests.

In light of the reviews, which you will find at the end of this email, we are pleased to offer you the opportunity to address the remaining points from the reviewers in a revision that we anticipate should not take you very long. We will then assess your revised manuscript and your response to the reviewers' comments with our Academic Editor aiming to avoid further rounds of peer-review, although might need to consult with the reviewers, depending on the nature of the revisions.

**IMPORTANT - SUBMITTING YOUR REVISION**

*Resubmission Checklist*

*Published Peer Review*

*PLOS Data Policy*

*Blot and Gel Data Policy*

Sincerely,

Roli Rpberts

Roland Roberts, PhD

Senior Editor

PLOS Biology

rroberts@plos.org

REVIEWERS' COMMENTS:

Reviewer #1:

I appreciated the authors' effort to address my comments. I am satisfied with their responses and have no further comments.

Reviewer #3:

[identifies himself as Diego Barneche]

I was invited to provide an additional independent assessment on the work by Audzijonyte et al. The authors used a physiologically structured food web model (a multi-species size spectrum model) to simulate how temperature and primary production from two different sources (benthic and planktonic) might affect a multitude of responses in the fish community, e.g., biomass, yield, average size of individuals. In particular, the authors were interested in evaluating the relative contribution of these variables to the responses based on a benchmark equilibrium state.

I have been given access to the previous round of reviews as well as the authors' rebuttal. I think the authors have done a good job at addressing the previous set of comments, and I agree with them that this is a timely contribution to the field. The model makes advances compared to previous ones, and it is (as much as realistically possible) well parametrised based on a unique long-term monitoring dataset from a region of relevance where warming effects have already been manifesting strongly. In my opinion, the key here is to make sure that the language reflects well the simulation nature of this study; and the authors certainly did a great job at clarifying that in this round. Moreover, the authors have provided an objective approach to evaluating parameter uncertainty, and have been explicit about the limitations of their approach (lines 459-465).

I do think though that some questions might be further explored in this paper. In my comments below, I will focus on some modelling questions (sensitivity analyses), overall suggestions to further clarify the text and to expand the discussion. I hope the authors find my comments useful.

# General comments:

## Abstract

Here, the language could be improved to avoid confusion and be more explicit about the simulation nature of the study. Words like "evaluate", "identify", "drivers" give the impression that this is a data-driven predictive model. I make specific suggestions below. The conclusion sentence between lines 40-42 is not novel nor surprising. To me, to last two sentences of the abstract highlight well the novelty and relevance of the study.

## Introduction

- Lines 127-129: This question 1) came (in part) as a surprise because the introduction does not articular how changes in food source will affect average size of individuals. Do you mean in similar ways as to a temperature-size rule effect? There's much better understanding of how temperature will affect size, but much less evidence with regards to the effect of overall food supply. This is generally thought to have a direct effect on overall population abundance, not average size of individuals. Whatever the mechanism, I think it would be important to articulate this expectation in the introduction.

## Methods:

- Maybe it is an issue of pdf resolution, but the version of fig 1 available to me makes it really hard to visualise the red circles on the map. Is there a better way to clarify this? Perhaps by adding a third zoomed layer?

- Lines 167-172: I would argue that some top predators can be overestimated in reef visual surveys on a per-unit-area basis. As in, they likely obtain their energy sources from an area that is much larger (100s km) than the relatively smaller, typical survey areas (100-500 m2). How did the authors deal with that? An explanation would be well placed between lines 167-172.

- Are lobsters and urchins being given unique importance because of commercial value, or because they are the main invertebrate prey in fish diets in Tasmania, or simply because they're the only invertebrate species that can be parametrised in terms of size? What would happen to the model outputs if urchins and lobsters were modelled as being part of the "background resource" like all other invertebrates? I think this is an important question that needs to be addressed in the appendix because it will speak to the robustness of simulation findings conditioned on model parametrisation an assumptions made for the base of the food chain.

- Is fishing mortality size dependent in the MSSS (I suspect not)? If not, a justification needs to be included because many fishery species have size-dependent fishing (as well as natural) mortality.

- Lines 358-362: I would encourage the authors to consider an alternative approach to inclusion of temperature effects on the model. Daily CMIP6-derived SST time series (RCP8.5) for the region can be downloaded from NCI-ESGF, at least until 2050 (https://pcmdi.llnl.gov/CMIP6/ArchiveStatistics/esgf_data_holdings/). If the idea is to create as much realism as possible, why not use that rather than a fixed end-point temperature prediction?

## Results & Discussion

- I cannot help but wonder what the high-level expectation of the proposed scenarios will be, i.e., completely ignoring all of the complex mechanisms that compose the MSSS. For example, if overall per-capita metabolic demand is to increase by ~20% (a consequence of the 0.63 eV and a +2.5˚C change in temperature), and resource abundance changes by X fold (changes with scenarios), then what is the naive expectation of overall change in community biomass (i.e. assume no change in size structure)? I guess this would be extremely useful to help the reader appreciate how much the added mechanisms in the MSSS influence the departure from such a naive expectation. If the overall ratio between autotrophic and heterotrophic biomass doesn't change in the model, then I would expect the model to yield results comparable to the naive expectation. This would make a really cool expansion of Figure 4. There could a 5th panel containing the community-level changes, and another dashed line showing the naive expectation.

- Lines 740-749: It would be good to state whether maximum size is present in the mizer framework. If so, is it allowed to evolve? I'm not sure if this component is captured in the explanation given between lines 765-784. Otherwise how are changes in mean size emerging? The authors' explanation, although consistent with empirical data (lines 750-764), seems to imply that baseline conditions wouldn't allow individuals to achieve their maximum potential (size wise). My understanding is that individuals generally do not achieve their maximum sizes because they die early (so it's more about time rather than resource abundance). Generally more food would allow more individuals over time, but not necessarily bigger individuals. Some discussion around the relevance of this to wider ecological literature would be helpful.

## Additional thoughts: 

- What happens if the model does not have the additional 5 kg predator?

- How does the model deal with detrital recycling? It can be a major source of energy in the system (e.g. Brandl et al. 2019 10.1126/science.aav3384).

- Are current conditions under equilibrium? There's 30 years worth of calibration data—what does this reveal in terms of equilibrium?

- The authors did a really good job at stating the expectations/generalisations from their a study while acknowledging major limitations. There are a few missing processes, though, which I think would be important to mention explicitly. For example, how are nutrient regimes expected to change over time in SE Tasmania (see, e.g., Moore et al. 2019 10.1126/science.aao6379)? This could have major implications to primary production. The last paragraph briefly mentions species reorganisation, but it does not discuss the potential effects of larval connectivity coming from the north. Finally, another important aspect would be species metabolic adaptation which can happen quite fast, particularly in phytoplankton, and the prognosis does not look encouraging (see e.g. Barton et al. 2020 10.1111/ele.13469). From their scenarios, which one(s) would be most consistent with these predictions of nutrient and yield in phytoplankton primary production? This would be an important aspect to highlight.

# Specific comments:

words in [] mean an addition, words in ** mean a deletion

## Overall

- Standardise the use of multi-species. In some parts of the text it has been written as "multispecies" (no hyphen).

## Abstract

- Line 30: "... specifically designed to *evaluate* [simulate] potential ..."

- Line 31: "... and *identify* [calculate] ..."

- Line 32: "... these *drivers* [variables] ..."

## Introduction

- Line 68: Consider citing De Roos & Persson (2001) here.

- Lines 89-92: This is an important statement that needs to be backed up by a reference.

- Line 103: "... *substrate* [substratum] ..."

- Line 112: "... *refugia* [refuge] ..."

## Methods

- Line 215: But was growth efficiency used in your model or not? This is likely a temperature dependent process, and theoretically said dependence might affect the slope of the size spectrum.

- Line 291: Is this macroalgal biomass density typical in SE Tasmania? This needs explicit mention in the text.

- Line 302: Please cite Yvon-Durocher et al. 2011 (10.1111/j.1365-2486.2010.02321.x)-this is data based evidence that warming steepens the size spectrum slope.

- Line 342: That equation is not actually used in Brown et al. 2004. It is proposed (equation 3) in Gillooly et al. 2001 (10.1126/science.1061967). Please swap the citation to the correct one.

- Line 347: Please provide a reference for the 0.63 eV choice. Marine phytoplankton can have much steeper temperature dependence than heterotrophic organisms, particularly those from cooler water regimes. Please see Barton & Yvon-Durocher 2019 (10.1002/lno.11170).

## Results & Discussion

- Fig 3: What are the units across the x axes?

- Lines 624-627: I would suggest removing this sentence entirely, it does not fit well with the ecological relevance of the finding, which is the main topic of the paragraph.

- Lines 627-629: I would cite Naumann et al. 2013 (10.1371/journal.pone.0082923), who suggest that benthic primary production can in fact be substantial. This helps the cause.

---

## [Editor Report · Decision Letter 4]

27 Sep 2023

Dear Dr Delius,

Thank you for your patience while we considered your revised manuscript "A new size-based model highlights the importance of benthic production changes for understanding climate change impacts on a temperate coastal ecosystem" for publication as a Research Article at PLOS Biology. This revised version of your manuscript has been evaluated by the PLOS Biology editors and the Academic Editor.

Based on our Academic Editor's assessment of your revision, we are likely to accept this manuscript for publication, provided you satisfactorily address the following data and other policy-related requests.

IMPORTANT - please attend to the following:

a) We think that the Title would be more appealing if the findings were put first (rather than the method) and if some of the specialised terms were unpacked a little. We suggest something like "Changes in sea floor productivity have a crucial role on the impact of climate change in temperate coastal ecosystems according to a new model that accounts for species size" or "Changes in sea floor productivity are crucial to understanding the impact of climate change in temperate coastal ecosystems according to a new size-based model" (you may have a suggestion that more accurately reflects the study, and we'd be happy to consider that)

b) Many thanks for providing the underlying data and code. Please could you cite the location of the data clearly in all relevant main and supplementary Figure legends (i.e. Figs 3, 4, S1, S2, S3 S4, S5, S6, S7, S8), e.g. “The data underlying this Figure can be found in https://doi.org/10.5281/zenodo.8281030"

We expect to receive your revised manuscript within two weeks. 

*Published Peer Review History*

*Press*

Sincerely,

Roli Roberts

Roland Roberts, PhD

Senior Editor,

rroberts@plos.org,

PLOS Biology

DATA NOT SHOWN?

---

## [Editor Report · Decision Letter 5]

19 Oct 2023

Dear Dr Delius,

Thank you for the submission of your revised Research Article "Changes in sea floor productivity are crucial to understanding the impact of climate change in temperate coastal ecosystems according to a new size-based model" for publication in PLOS Biology. On behalf of my colleagues and the Academic Editor, Michel Loreau, I'm pleased to say that we can in principle accept your manuscript for publication, provided you address any remaining formatting and reporting issues. These will be detailed in an email you should receive within 2-3 business days from our colleagues in the journal operations team; no action is required from you until then. Please note that we will not be able to formally accept your manuscript and schedule it for publication until you have completed any requested changes.

Sincerely, 

Roli Roberts

Senior Editor

PLOS Biology

rroberts@plos.org